# Sparse Bayesian structure learning with dependent relevance determination prior

**Anqi Wu**[1]    **Mijung Park**[2]    **Oluwasanmi Koyejo**[3]    **Jonathan W. Pillow**[4]

[1,4] Princeton Neuroscience Institute, Princeton University,
{`anqiw, pillow`}`@princeton.edu`
[2] The Gatsby Unit, University College London, `mijung@gatsby.ucl.ac.uk`
[3] Department of Psychology, Stanford University, `sanmi@stanford.edu`

## Abstract

In many problem settings, parameter vectors are not merely sparse, but dependent in such a way that non-zero coefficients tend to cluster together. We refer to this form of dependency as "region sparsity". Classical sparse regression methods, such as the lasso and automatic relevance determination (ARD), model parameters as independent *a priori*, and therefore do not exploit such dependencies. Here we introduce a hierarchical model for smooth, region-sparse weight vectors and tensors in a linear regression setting. Our approach represents a hierarchical extension of the relevance determination framework, where we add a transformed Gaussian process to model the dependencies between the prior variances of regression weights. We combine this with a structured model of the prior variances of Fourier coefficients, which eliminates unnecessary high frequencies. The resulting prior encourages weights to be region-sparse in two different bases simultaneously. We develop efficient approximate inference methods and show substantial improvements over comparable methods (e.g., group lasso and smooth RVM) for both simulated and real datasets from brain imaging.

## 1 Introduction

Recent work in statistics has focused on high-dimensional inference problems where the number of parameters $p$ equals or exceeds the number of samples $n$. Although ill-posed in general, such problems are made tractable when the parameters have special structure, such as sparsity in a particular basis. A large literature has provided theoretical guarantees about the solutions to sparse regression problems and introduced a suite of practical methods for solving them efficiently [1–7].

The Bayesian interpretation of standard "shrinkage" based methods for sparse regression problems involves maximum a postieriori (MAP) inference under a sparse, independent prior on the regression coefficients [8–15]. Under such priors, the posterior has high concentration near the axes, so the posterior maximum is at zero for many weights unless it is pulled strongly away by the likelihood. However, these independent priors neglect a statistical feature of many real-world regression problems, which is that non-zero weights tend to arise in clusters, and are therefore not independent *a priori*. In many settings, regression weights have an explicit topographic relationship, as when they index regressors in time or space (e.g., time series regression, or spatio-temporal neural receptive field regression). In such settings, nearby weights exhibit dependencies that are not captured by independent priors, which results in sub-optimal performance.

Recent literature has explored a variety of techniques for improving sparse inference methods by incorporating different types of prior dependencies, which we will review here briefly. The *smooth relevance vector machine* (s-RVM) extends the RVM to incorporate a smoothness prior defined

in a kernel space, so that weights are smooth as well as sparse in a particular basis [16]. The *group lasso* captures the tendency for groups of coefficients to remain in or drop out of a model in a coordinated manner by using an $l_1$ penalty on the $l_2$ norms pre-defined groups of coefficients [17]. A method described in [18] uses a multivariate Laplace distribution to impose spatio-temporal coupling between prior variances of regression coefficients, which imposes group sparsity while leaving coefficients marginally uncorrelated. The literature includes many related methods [19–24], although most require *a priori* knowledge of the dependency structure, which may be unavailable in many applications of interest.

Here we introduce a novel, flexible method for capturing dependencies in sparse regression problems, which we call dependent relevance determination (DRD). Our approach uses a Gaussian process to model dependencies between latent variables governing the prior variance of regression weights. (See [25], which independently proposed a similar idea.) We simultaneously impose smoothness by using a structured model of the prior variance of the weights' Fourier coefficients. The resulting model captures sparse, local structure in two different bases simultaneously, yielding estimates that are sparse as well as smooth. Our method extends previous work on automatic locality determination (ALD) [26] and Bayesian structure learning (BSL) [27], both of which described hierarchical models for capturing sparsity, locality, and smoothness. Unlike these methods, DRD can tractably recover region-sparse estimates with multiple regions of non-zero coefficients, without pre-defining number of regions. We argue that DRD can substantially improve structure recovery and predictive performance in real-world applications.

This paper is organized as follows: Sec. 2 describes the basic sparse regression problem; Sec. 3 introduces the DRD model; Sec. 4 and Sec. 5 describe the approximate methods we use for inference; In Sec. 6, we show applications to simulated data and neuroimaging data.

## 2    Problem setup

### 2.1    Observation model

We consdier a scalar response $y_i \in \mathbb{R}$ linked to an input vector $\mathbf{x}_i \in \mathbb{R}^p$ via the linear model:

$$y_i = \mathbf{x}_i^\top \mathbf{w} + \epsilon_i, \quad \text{for} \quad i = 1, 2, \cdots, n, \tag{1}$$

with observation noise $\epsilon_i \sim \mathcal{N}(0, \sigma^2)$. The regression (linear weight) vector $\mathbf{w} \in \mathbb{R}^p$ is the quantity of interest. We denote the design matrix by $X \in \mathbb{R}^{n \times p}$, where each row of $X$ is the $i^{th}$ input vector $\mathbf{x}_i^\top$, and the observation vector by $\mathbf{y} = [y_1, \cdots, y_n]^\top \in \mathbb{R}^n$. The likelihood can be written:

$$\mathbf{y}|X, \mathbf{w}, \sigma^2 \sim \mathcal{N}(\mathbf{y}|X\mathbf{w}, \sigma^2 I). \tag{2}$$

### 2.2    Prior on regression vector

We impose the zero-mean multivariate normal prior on $\mathbf{w}$:

$$\mathbf{w}|\boldsymbol{\theta} \sim \mathcal{N}(0, C(\boldsymbol{\theta})) \tag{3}$$

where the prior covariance matrix $C(\boldsymbol{\theta})$ is a function of hyperparameters $\boldsymbol{\theta}$. One can specify $C(\boldsymbol{\theta})$ based on prior knowledge on the regression vector, e.g. sparsity [28–30], smoothness [16, 31], or both [26]. Ridge regression assumes $C(\theta) = \theta^{-1} I$ where $\theta$ is a scalar for precision. Automatic relevance determination (ARD) uses a diagonal prior covariance matrix with a distinct hyperparameter $\theta_i$ for each element of the diagonal, thus $C_{ii} = \theta_i^{-1}$. Automatic smoothness determination (ASD) assumes a non-diagonal prior covariance, given by a Gaussian kernel, $C_{ij} = \exp(-\rho - \Delta_{ij}/2\delta^2)$ where $\Delta_{ij}$ is the squared distance between the filter coefficients $\mathbf{w}_i$ and $\mathbf{w}_j$ in pixel space and $\boldsymbol{\theta} = \{\rho, \delta^2\}$. Automatic locality determination (ALD) parametrizes the local region with a Gaussian form, so that prior variance of each filter coefficient is determined by its Mahalanobis distance (in coordinate space) from some mean location $\nu$ under a symmetric positive semi-definite matrix $\Psi$. The diagonal prior covariance matrix is given by $C_{ii} = \exp(-\frac{1}{2}(\chi_i - \nu)^\top \Psi^{-1}(\chi_i - \nu)))$, where $\chi_i$ is the space-time location (i.e., filter coordinates) of the $i^{th}$ filter coefficient $\mathbf{w}_i$ and $\boldsymbol{\theta} = \{\nu, \Psi\}$.

## 3   Dependent relevance determination (DRD) priors

We formulate the prior covariances to capture the region dependent sparsity in the regression vector in the following.

**Sparsity inducing covariance**

We first parameterise the prior covariance to capture region sparsity in $\mathbf{w}$

$$C_s = \mathrm{diag}[\exp(\mathbf{u})], \tag{4}$$

where the parameters are $\mathbf{u} \in \mathbb{R}^p$. We impose the Gaussian process (GP) hyperprior on $\mathbf{u}$

$$\mathbf{u} \sim \mathcal{N}(b\mathbf{1}, K). \tag{5}$$

The GP hyperprior is controlled by the mean parameter $b \in \mathbb{R}$ and the squared exponential kernel parameters, overall scale $\rho \in \mathbb{R}$ and the length scale $l \in \mathbb{R}$. We denote the hyperparameters by $\boldsymbol{\theta}_s = \{b, \rho, l\}$. We refer to the prior distribution associated with the covariance $C_s$ as *dependent relevance determination* (DRD) prior.

Note that this hyperprior induces dependencies between the ARD precisions, that is, prior variance changes slowly between neighboring coefficients. If the $i^{th}$ coefficient of $\mathbf{u}$ has large prior variance, then probably the $i+1$ and $i-1$ coefficients are large as well.

**Smoothness inducing covariance**

We then formulate the smoothness inducing covariance in frequency domain. Smoothness is captured by a low pass filter with only lower frequencies passing through. Therefore, we define a zero-mean Gaussian prior over the Fourier-transformed weights $\mathbf{w}$ using a diagonal covariance matrix $C_f$ with diagonal

$$C_{f,ii} = \exp(-\frac{\chi_i^2}{2\delta^2}), \tag{6}$$

where $\chi_i$ is the $i^{th}$ location of the regression weights $\mathbf{w}$ in frequency domain and $\delta^2$ is the Gaussian covariance. We denote the hyperparameters by $\boldsymbol{\theta}_f = \delta^2$. This formulation imposes neighboring weights to have similar levels of Fourier power.

Similar to the automatic determination in frequency coordinates (ALDf) [26], this way of formulating the covariance requires taking discrete Fourier transform of input vectors to construct the prior in the frequency domain. This is a significant consumption in computation and memory requirements especially when $p$ is large. To avoid the huge expense, we abandon the single frequency version $C_f$ but combine it with $C_s$ to form $C_{sf}$ with both sparsity and smoothness induced as the following.

**Smoothness and region sparsity inducing covariance**

Finally, to capture both region sparsity and smoothness in $\mathbf{w}$, we combine $C_s$ and $C_f$ in the following way

$$C_{sf} = C_s^{\frac{1}{2}} B^\top C_f B C_s^{\frac{1}{2}}, \tag{7}$$

where $B$ is the Fourier transformation matrix which could be huge when $p$ is large. Implementation exploits the speed of the FFT to apply B implicitly. This formulation implies that the sparse regions that are captured by $C_s$ are pruned out and the variance of the remaining entries in weights are correlated by $C_f$. We refer to the prior distribution associated with the covariance $C_{sf}$ as *smooth dependent relevance determination* (sDRD) prior.

Unlike $C_s$, the covariance $C_{sf}$ is no longer diagonal. To reduce computational complexity and storage requirements, we only store those values that correspond to non-zero portions in the diagonal of $C_s$ and $C_f$ from the full $C_{sf}$.

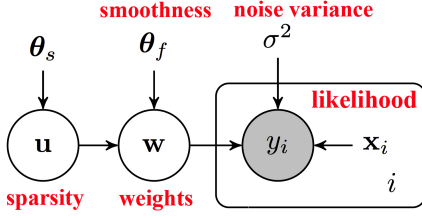

Figure 1: Generative model for locally smooth and globally sparse Bayesian structure learning. The $i^{th}$ response $y_i$ is linked to an input vector $\mathbf{x}_i$ and a weight vector $\mathbf{w}$ in each $i$. The weight vector $\mathbf{w}$ is governed by $\mathbf{u}$ and $\boldsymbol{\theta}_f$. The hyper-prior $p(\mathbf{u}|\boldsymbol{\theta}_s)$ imposes correlated sparsity in $\mathbf{w}$ and the hyperparameters $\boldsymbol{\theta}_f$ imposes smoothness in $\mathbf{w}$.

## 4 Posterior inference for $\mathbf{w}$

First, we denote the overall hyperparameter set to be $\boldsymbol{\theta} = \{\sigma^2, \boldsymbol{\theta}_s, \boldsymbol{\theta}_f\} = \{\sigma^2, b, \rho, l, \delta^2\}$. We compute the maximum likelihood estimate for $\boldsymbol{\theta}$ (denoted by $\hat{\boldsymbol{\theta}}$) and compute the conditional MAP estimate for the weights $\mathbf{w}$ given $\hat{\boldsymbol{\theta}}$ (closed form), which is the *empirical Bayes* procedure equipped with a hyper-prior. Our goal is to infer $\mathbf{w}$. The posterior distribution over $\mathbf{w}$ is obtained by

$$p(\mathbf{w}|X, \mathbf{y}) = \int\int p(\mathbf{w}, \mathbf{u}, \boldsymbol{\theta}|X, \mathbf{y})d\mathbf{u}d\boldsymbol{\theta}, \tag{8}$$

which is analytically intractable. Instead, we approximate the marginal posterior distribution with the conditional distribution given the MAP estimate of $\mathbf{u}$, denoted by $\boldsymbol{\mu}_\mathbf{u}$, and the maximum likelihood estimation of $\sigma^2, \boldsymbol{\theta}_s, \boldsymbol{\theta}_f$ denoted by $\hat{\sigma^2}, \hat{\boldsymbol{\theta}}_s, \hat{\boldsymbol{\theta}}_f$,

$$p(\mathbf{w}|X, \mathbf{y}) \approx p(\mathbf{w}|X, \mathbf{y}, \boldsymbol{\mu}_\mathbf{u}, \hat{\sigma^2}, \hat{\boldsymbol{\theta}}_s, \hat{\boldsymbol{\theta}}_f). \tag{9}$$

The approximate posterior over $\mathbf{w}$ is multivariate normal with the mean and covariance given by

$$p(\mathbf{w}|X, \mathbf{y}, \boldsymbol{\mu}_\mathbf{u}, \hat{\sigma^2}, \hat{\boldsymbol{\theta}}_s, \hat{\boldsymbol{\theta}}_f) = \mathcal{N}(\boldsymbol{\mu}_\mathbf{w}, \Lambda_\mathbf{w}), \tag{10}$$

$$\Lambda_\mathbf{w} = (\frac{1}{\hat{\sigma^2}}X^\top X + C^{-1}_{\boldsymbol{\mu}_\mathbf{u}, \hat{\boldsymbol{\theta}}_s, \hat{\boldsymbol{\theta}}_f})^{-1}, \tag{11}$$

$$\boldsymbol{\mu}_\mathbf{w} = \frac{1}{\hat{\sigma^2}}\Lambda_\mathbf{w}X^T\mathbf{y}. \tag{12}$$

## 5 Inference for hyperparameters

The MAP inference of $\mathbf{w}$ derived in the previous section depends on the values of $\hat{\boldsymbol{\theta}} = \{\hat{\sigma^2}, \hat{\boldsymbol{\theta}}_s, \hat{\boldsymbol{\theta}}_f\}$. To estimate $\hat{\boldsymbol{\theta}}$, we maximize the marginal likelihood of the evidence.

$$\hat{\boldsymbol{\theta}} = \arg\max_{\boldsymbol{\theta}} \log p(\mathbf{y}|X, \boldsymbol{\theta}) \tag{13}$$

where

$$p(\mathbf{y}|X, \boldsymbol{\theta}) = \int\int p(\mathbf{y}|X, \mathbf{w}, \sigma^2)p(\mathbf{w}|\mathbf{u}, \boldsymbol{\theta}_f)p(\mathbf{u}|\boldsymbol{\theta}_s)d\mathbf{w}d\mathbf{u}. \tag{14}$$

Unfortunately, computing the double integrals is intractable. In the following, we consider the the approximation method based on Laplace approximation to compute the integral approximately.

**Laplace approximation to posterior over $\mathbf{u}$**

To approximate the marginal likelihood, we can rewrite Bayes' rule to express the marginal likelihood as the likelihood times prior divided by the posterior,

$$p(\mathbf{y}|X, \boldsymbol{\theta}) = \frac{p(\mathbf{y}|X, \mathbf{u})p(\mathbf{u}|\boldsymbol{\theta})}{p(\mathbf{u}|\mathbf{y}, X, \boldsymbol{\theta})}, \tag{15}$$

Laplace's method allows us to approximate $p(\mathbf{u}|\mathbf{y}, X, \boldsymbol{\theta})$, the posterior over the latent $\mathbf{u}$ given the data $\{X, \mathbf{y}\}$ and hyper-parameters $\boldsymbol{\theta}$, using a Gaussian centered at the mode of the distribution and inverse covariance given by the Hessian of the negative log-likelihood. Let $\boldsymbol{\mu}_\mathbf{u} = \arg\max_\mathbf{u} p(\mathbf{u}|\mathbf{y}, X, \boldsymbol{\theta})$ and $\Lambda_\mathbf{u} = -(\frac{\partial^2}{\partial\mathbf{u}\partial\mathbf{u}^\top}\log p(\mathbf{u}|\mathbf{y}, X, \boldsymbol{\theta}))^{-1}$ denote the mean and covariance

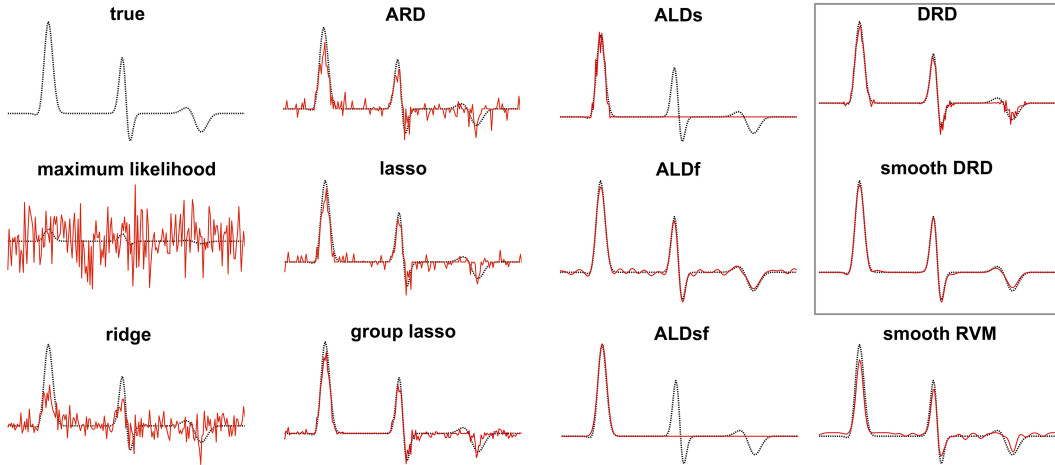

Figure 2: Comparison of estimators for 1D simulated example. **First column**: True filter weight, maximum likelihood (linear regression) estimate, empirical Bayesian ridge regression (L2-penalized) estimate; **Second column**: ARD estimate with different and independent prior covariance hyperparameters, lasso regression with L1-regularization and group lasso with group size of 5; **Third column**: ALD methods in space-time domain, frequency domain and combination of both, respectively; **Fourth column**: DRD method in space-time domain only and its smooth version sDRD imposing both sparsity (space-time) and smoothness (frequency), and smooth RVM initialized with elastic net estimate.

of this Gaussian, respectively. Although the right-hand-side can be evaluated at any value of $\mathbf{u}$, a common approach is to use the mode $\mathbf{u} = \boldsymbol{\mu}_{\mathbf{u}}$, since this is where the Laplace approximation is most accurate. This leads to the following expression for the log marginal likelihood:

$$\log p(\mathbf{y}|X, \boldsymbol{\theta}) \quad \approx \quad \log p(\mathbf{y}|X, \boldsymbol{\mu}_{\mathbf{u}}) + \log p(\boldsymbol{\mu}_{\mathbf{u}}|\boldsymbol{\theta}) - \tfrac{1}{2} \log |2\pi \Lambda_{\mathbf{u}}|. \tag{16}$$

Then by optimizing $\log p(\mathbf{y}|X, \boldsymbol{\theta})$ with regard to $\boldsymbol{\theta}$, we can get $\hat{\boldsymbol{\theta}}$ given a fixed $\boldsymbol{\mu}_{\mathbf{u}}$, denoted as $\hat{\boldsymbol{\theta}}_{\boldsymbol{\mu}_{\mathbf{u}}}$. Following an iterative optimization procedure, we obtain an updating rule $\boldsymbol{\mu}_{\mathbf{u}}^t = \arg\max_{\mathbf{u}} p(\mathbf{u}|\mathbf{y}, X, \hat{\boldsymbol{\theta}}_{\boldsymbol{\mu}_{\mathbf{u}}^{t-1}})$ at $t^{th}$ iteration. The algorithm will stop when $\mathbf{u}$ and $\boldsymbol{\theta}$ converge. More information and details about formulation and derivation are described in the appendix.

## 6 Experiment and Results

### 6.1 One Dimensional Simulated Data

Beginning with simulated data, we compare performances of various regression estimators. One dimensional data is generated from a generative model with $d = 200$ dimensions. Firstly to generate a Gaussian process, a covariance kernel matrix $K$ is built with squared exponential kernel with the spatial locations of regression weights as inputs. Then a scalar $b$ is set as the mean function to determine the scale of prior covariance. Given the Gaussian process, we generate a multivariate vector $\mathbf{u}$, and then take its exponential to obtain the diagonal of prior covariance $C_s$ in space-time domain. To induce smoothness, eq. 7 is introduced to get covariance $C_{sf}$. Then a weight vector $\mathbf{w}$ is sampled from a Gaussian distribution with zero mean and $C_{sf}$. Finally, we obtain the response $\mathbf{y}$ given stimulus $\mathbf{x}$ with $\mathbf{w}$ plus Gaussian noise $\epsilon$. In our case, $\epsilon$ should be large enough to ensure that data and response won't impose strong likelihood over prior knowledge. Thus the introduced prior would largely dominate the estimate. Three local regions are constructed, which are positive, negative and a half-positive-half-negative with sufficient zeros between separate bumps clearly apart. As shown in Figure 2, the left top subfigure shows the underlying weight vector $\mathbf{w}$.

Traditional methods like maximum likelihood, without any prior, are significantly overwhelmed by large noise of the data. Weak priors such as ridge, ARD, lasso could fit the true weight better with

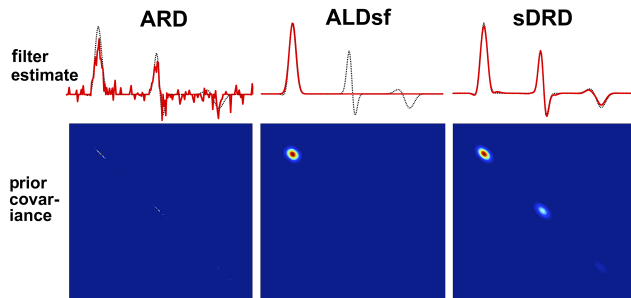

Figure 3: Estimated filter weights and prior covariances. **Upper row** shows the true filter (dotted black) and estimated ones (red); **Bottom row** shows the underlying prior covariance matrix.

different levels of sparsity imposed, but are still not sparse enough and not smooth at all. Group lasso enforces a stronger sparsity than lasso by assuming block sparsity, thus making the result smoother locally. ALD based methods have better performance, compared with traditional ones, in identifying one big bump explicitly. ALDs is restricted by the assumption of one modal Gaussian, therefore is able to find one dominating local region. ALDf focuses localities in frequency domain thus make the estimate smoother but no spatial local regions are discovered. ALDsf combines the effects in both ALDs and ALDf, and thus possesses smoothness but only one region is found. Smooth Relevance Vector Machine (sRVM) can smooth the curve by incorporating a flexible noise-dependent smoothness prior into the RVM, but is not able to draw information from data likelihood magnificently. Our DRD can impose distinct local sparsity via Gaussian process prior and sDRD can induce smoothness via bounding the frequencies. For all baseline models, we do model selection via cross-validation varying through a wide range of parameter space, thus we can guarantee the fairness for comparisons.

To further illustrate the benefits and principles of DRD, we demonstrate the estimated covariance via ARD, ALDsf and sDRD in Figure 3. It can be stated that ARD could detect multiple localities since its priors are purely independent scalars which could be easily influenced by data with strong likelihood, but the consideration is the loss of dependency and smoothness. ALDsf can only detect one locality due to its deterministic Gaussian form when likelihood is not sufficiently strong, but with Fourier components over the prior, it exhibits smoothness. sDRD could capture multiple local sparse regions as well as impose smoothness. The underlying Gaussian process allows multiple non-zero regions in prior covariance with the result of multiple local sparsities for weight tensor. Smoothness is introduced by a Gaussian type of function controlling the frequency bandwidth and direction.

In addition, we examine the convergence properties of various estimators as a function of the amount of collected data and give the average relative errors of each method in Figure 4. Responses are simulated from the same filter as above with large Gaussian white noise which weakens the data likelihood and thus guarantees a significant effect of prior over likelihood. The results show that sDRD estimate achieves the smallest MSE (mean squared error), regardless of the number of training samples. The MSE, mentioned here and in the following paragraphs, refers to the error compared with the underlying $\mathbf{w}$. The test error, which will be mentioned in later context, refers to the error compared with true $\mathbf{y}$. The left plot in Figure 4 shows that other methods require at least 1-2 times more data than sDRD to achieve the same error rate. The right figure shows the ratio of the MSE for each estimate to the MSE for sDRD estimate, showing that the next best method (ALDsf) exhibits an error nearly two times of sDRD.

## 6.2 Two Dimensional Simulated Data

To better illustrate the performance of DRD and lay the groundwork for real data experiment, we present a 2-dimensional synthetic experiment. The data is generated to match characteristics of real fMRI data, as will be outlined in the next section. With a similar generation procedure as in 1-dimensional experiment, a 2-dimensional $\mathbf{w}$ is generated with analogical properties as the regression weights in fMRI data. The analogy is based on reasonable speculation and accumulated acknowledge from repeated trials and experiment. Two comparative studies are conducted to investigate the influences of sample size on the recovery accuracy of $\mathbf{w}$ and predictive ability, both with dimension = 1600 (the same as fMRI). To demonstrate structural sparsity recovery, we only compare our DRD method with ARD, lasso, elastic net (elnet), group lasso (glasso).

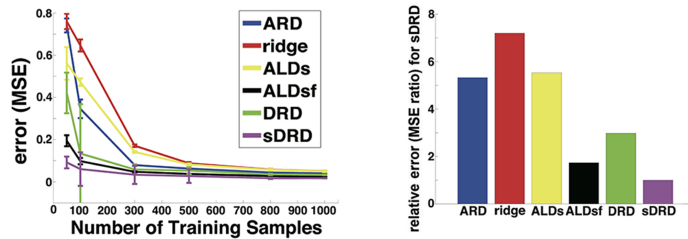

Figure 4: Convergence of error rates on simulated data with varying training size (**Left**) and the relative error (MSE ratio) for sDRD (**Right**)

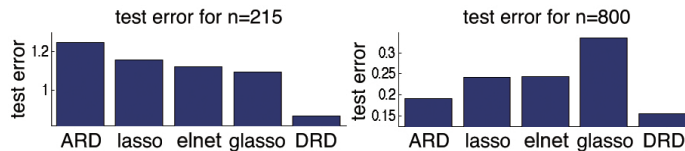

Figure 5: Test error for each method when $n = 215$ (**Left**) and $n = 800$ (**Right**) for 2D simulated data.

The sample size $n$ varies in $\{215, 800\}$. The results are shown in Fig. 5 and Fig. 6. When $n = 215$, only DRD is able to reveal an approximative estimation of true $\mathbf{w}$ with a small level of noise as well as giving the lowest predictive error. Group lasso performs slightly better than ARD, lasso and elnet, and presents only a weakly distinct block wise estimation compared with lasso and elnet. Lasso and elnet both show similar performances and give a stronger sparsity than ARD, which indicates that ARD fails to impose strong sparsity in this synthetic case due to its complete independencies among dimensions when data is less sufficient and noisy. When $n = 800$, DRD still holds the best prediction. Group lasso fails to keep the record since block-wise penalty can capture group information but miss the subtlety when finer details matter. ARD progresses to the second place because when data likelihood is strong enough, posterior of $\mathbf{w}$ won't be greatly influenced by the noise but follow the likelihood and the prior. Additionally, since ARD's prior is more flexible and independent than lasso and elnet, the posterior would approximate the underlying $\mathbf{w}$ better and finer.

### 6.3  fMRI Data

We analyzed functional MRI data from the Human Connectome Project [1] collected from 215 healthy adult participants on a relational reasoning task. We used contrast images for the comparison of relational reasoning and matching tasks. Data were processed using the HCP minimal preprocessing pipelines [32], down-sampled to $63 \times 76 \times 63$ voxels using the flirt applyXfm tool [33], then tailored further down to $40 \times 76 \times 40$ by deleting zero-signal regions outside the brain. We analyzed 215 samples, each of which is an average from Z-slice 37 to 39 slices of 3D structure based on recommendations by domain experts. As the dependent variable in the regression, we selected the number of correct responses on the Penn Matrix Text, which is a measure of fluid intelligence that should be related to relational reasoning performance.

In each run, we randomly split the fMRI data into five sets for five-fold cross-validation, and took an average of test errors across five folds for each run. Hyperparameters were chosen by a five-fold cross-validation within the training set, and the optimal hyper parameter set was used for computing test performance. Fig. 7 shows the regions of positive (red) and negative (blue) support for the regression weights we obtained using different sparse regression methods. The rightmost panel quantifies performance using mean squared error on held out test data. Both predictive performance and estimated pattern are similar to $n = 215$ result in 2D synthetic experiment. ARD returns a quite noisy estimation due to the complete independencies and weak likelihood. The elastic net estimate improves slightly over lasso but is significantly better than ARD, which indicates that lasso type of regularizations impose stronger sparsity than ARD in this case. Group lasso is slightly better

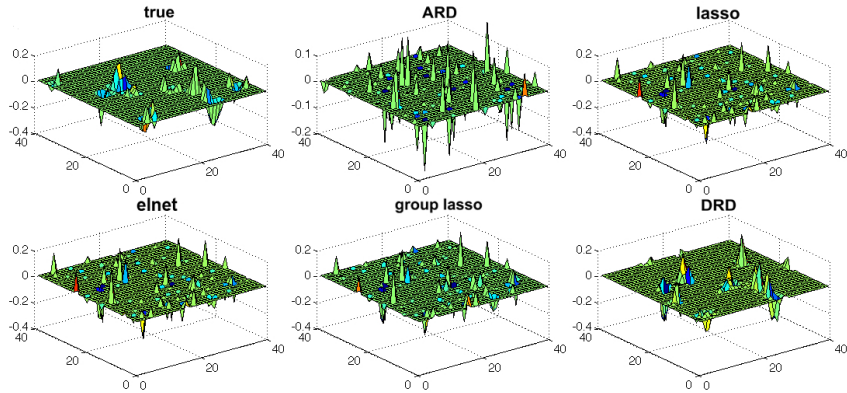

Figure 6: Surface plot of estimated $\mathbf{w}$ from each method using 2-dimensional simulated data when $n = 215$.

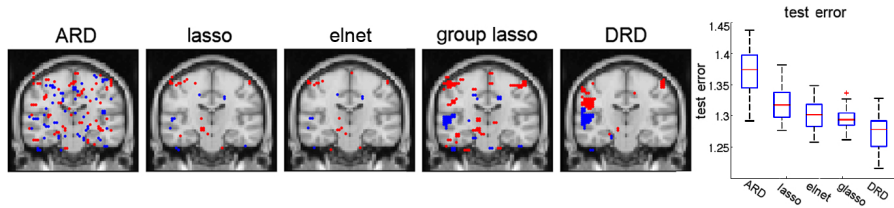

Figure 7: Positive (red) and negative (blue) supports of the estimated weights from each method using real fMRI data and the corresponding test errors.

because of its block-wise regularization, but more noisy blocks pop up influencing the predictive ability. DRD reveals strong sparsity as well as clustered local regions. It also possesses the smallest test error indicating the best predictive ability. Given that local group information most likely gather around a few pixels in fMRI data, smoothness would be less valuable to be induced. This is the reason that sDRD doesn't show a distinct outperforming result over DRD, as a result of which we omit smoothness imposing comparative experiment for fMRI data. In addition, we also test the StructOMP [24] method for both 2D simulated data and fMRI data, but it doesn't show satisfactory estimation and predictive ability in the 2D data with our data's intrinsic properties. Therefore we chose to not show it for comparison in this study.

# 7 Conclusion

We proposed DRD, a hierarchal model for smooth and region-sparse weight tensors, which uses a Gaussian process to model spatial dependencies in prior variances, an extension of the relevance determination framework. To impose smoothness, we also employed a structured model of the prior variances of Fourier coefficients, which allows for pruning of high frequencies. Due to the intractability of marginal likelihood integration, we developed an efficient approximate inference method based on Laplace approximation, and showed substantial improvements over comparable methods on both simulated and fMRI real datasets. Our method yielded more interpretable weights and indeed discovered multiple sparse regions that were not detected by other methods. We have shown that DRD can gracefully incorporate structured dependencies to recover smooth, region-sparse weights without any specification of groups or regions, and believe it will be useful for other kinds of high-dimensional datasets from biology and neuroscience.

### Acknowledgments

This work was supported by the McKnight Foundation (JP), NSF CAREER Award IIS-1150186 (JP), NIMH grant MH099611 (JP) and the Gatsby Charitable Foundation (MP).

## Footnotes

[1] http://www.humanconnectomeproject.org/.

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
