[Supplementary Material · Appendix for Sparse Bayesian structure learning with “dependent relevance determination” priors.pdf]

# Appendix

## A  Large Scale and Numerical Instability Issue

To deal with large scale data and avoid numerical instabilities, we deploy several tricks to speed up our implementation.

- First, we use incomplete Cholesky factorization [1] to obtain a low rank decomposition, then apply singular value decomposition (SVD) to the low rank matrix to fast obtain a stable inverse of kernel matrix, which is singular and therefore close to lowrank. After SVD, we project $\mathbf{u}$ and $b$ down to the lower dimension, which also reduces computational complexity.

- Secondly, we don't store Fourier transformation matrix $B$ but use fft() function in MATLAB to directly derive the Fourier transformation.

- Thirdly, based on the observation that the prior covariance matrix is extremely sparse or smooth, we trim all input dimensions by removing those corresponding to the small values of $C_s$ and $C_f$.

- Moreover, we formulate the objective in a more delicate way so that the largest rank of matrices in $\log p(\mathbf{u}|\mathbf{y}, X, \boldsymbol{\theta})$ is upper bounded by $\min(rank(C_s), rank(C_f))$, which is extremely small when the weight space is very smooth. Therefore when either is sparse enough, the entire computation would be fast and efficient.

  With these methods, the algorithm becomes scalable for large scale and high dimensional datasets.

## B  Derivatives of sDRD Using Primal Formulation

By Bayes' rule the posterior over the latent variable $\mathbf{u}$ is given by $p(\mathbf{u}|\mathbf{y}, X, \boldsymbol{\theta}) = p(\mathbf{u}|\boldsymbol{\theta})p(\mathbf{y}|X, \mathbf{u}, \boldsymbol{\theta})/p(\mathbf{y}|X, \boldsymbol{\theta})$, but as $p(\mathbf{y}|X, \boldsymbol{\theta})$ is independent of $\mathbf{u}$, we need only to consider the un-normalized posterior when maximizing w.r.t. $\mathbf{u}$. Similarly, maximizing $p(\mathbf{y}|X, \boldsymbol{\theta}) = p(\boldsymbol{\mu}_{\mathbf{u}}|\boldsymbol{\theta})p(\mathbf{y}|\boldsymbol{\mu}_{\mathbf{u}}, X, \boldsymbol{\theta})/p(\boldsymbol{\mu}_{\mathbf{u}}|\mathbf{y}, X)$ with respect to $\boldsymbol{\theta}$ equals to maximizing $p(\boldsymbol{\mu}_{\mathbf{u}}|\boldsymbol{\theta})p(\mathbf{y}|\boldsymbol{\mu}_{\mathbf{u}}, X, \boldsymbol{\theta})$. Taking the logarithm of $p(\mathbf{u}|\mathbf{y}, X, \boldsymbol{\theta})$ and using the primal formulation gives

$$\log p(\mathbf{u}|\mathbf{y}, X, \boldsymbol{\theta}) = \log p(\mathbf{y}|X, \mathbf{u}, \boldsymbol{\theta}_f, \sigma^2) + \log p(\mathbf{u}|\boldsymbol{\theta}_s) + const \tag{1}$$

$$\log p(\mathbf{y}|X, \mathbf{u}, \boldsymbol{\theta}_f, \sigma^2) = \log \int p(\mathbf{y}|X, \mathbf{w}, \sigma^2)p(\mathbf{w}|\mathbf{u}, \boldsymbol{\theta}_f)d\mathbf{w}$$

$$= -\frac{1}{2}\log|C_{sf}\Lambda^{-1}| + \frac{1}{2}\mu^\top\Lambda^{-1}\mu - \frac{1}{2\sigma^2}\mathbf{y}^\top\mathbf{y} - \frac{n}{2}\log\sigma^2 \tag{2}$$

$$\log p(\mathbf{u}|\boldsymbol{\theta}_s) = -\frac{1}{2}(\mathbf{u}-b\mathbf{1})^\top K_{\rho,l}^{-1}(\mathbf{u}-b\mathbf{1}) - \frac{1}{2}\log|2\pi K_{\rho,l}| \tag{3}$$

where $\Lambda = (\frac{1}{\sigma^2}X^\top X + C_{sf}^{-1})^{-1}$ and $\mu = \frac{1}{\sigma^2}\Lambda X^\top \mathbf{y}$ and $C_{sf} = C_s^{\frac{1}{2}}B^\top C_f BC_s^{\frac{1}{2}}$. $K_{\rho,l}$ is the covariance kernel function which is squared exponential function in our implementation,

$$K_{i,j} = \exp(-\frac{(\chi_i - \chi_j)^2}{2l^2} + \rho) \tag{4}$$

For efficient computation and restricting the dimension for matrix manipulation, we rearrange $\Lambda$ in the following way:

$$\Lambda = C_s^{\frac{1}{2}}(\frac{1}{\sigma^2}C_s^{\frac{1}{2}}X^\top XC_s^{\frac{1}{2}} + B^\top C_f^{-1}B)^{-1}C_s^{\frac{1}{2}}$$

$$= C_s^{\frac{1}{2}}B^\top C_f^{\frac{1}{2}}B(\frac{1}{\sigma^2}B^\top C_f^{\frac{1}{2}}BC_s^{\frac{1}{2}}X^\top XC_s^{\frac{1}{2}}B^\top C_f^{\frac{1}{2}}B + I)^{-1}B^\top C_f^{\frac{1}{2}}BC_s^{\frac{1}{2}} \tag{5}$$

where $I$ is an identity matrix. Let $Z = XC_s^{\frac{1}{2}}B^\top C_f^{\frac{1}{2}}B$, then the quadratic term is

$$\frac{1}{2}\mu^\top\Lambda^{-1}\mu = \frac{1}{2\sigma^4}\mathbf{y}^\top X(\frac{1}{\sigma^2}X^\top X + C_{sf}^{-1})^{-1}X^\top \mathbf{y}$$

$$= \frac{1}{2\sigma^4}\mathbf{y}^\top Z(\frac{1}{\sigma^2}Z^\top Z + I)^{-1}Z^\top \mathbf{y} \tag{6}$$

the log determinant term is

$$\frac{1}{2}\log|C_{sf}\Lambda^{-1}| = \frac{1}{2}\log|\frac{1}{\sigma^2}C_{sf}X^\top X + I| = \frac{1}{2}\log|\frac{1}{\sigma^2}Z^\top Z + I| \tag{7}$$

Thus

$$
\begin{aligned}
\log p(\mathbf{u}|\mathbf{y}, X, \boldsymbol{\theta}) &= -\frac{1}{2}\log|S| + \frac{1}{2\sigma^2}\mathbf{y}^\top Z S^{-1} Z^\top \mathbf{y} - \frac{1}{2\sigma^2}\mathbf{y}^\top \mathbf{y} \\
&\quad -\frac{1}{2}(\mathbf{u}-b\mathbf{1})^\top K_{\rho,l}^{-1}(\mathbf{u}-b\mathbf{1}) - \frac{1}{2}\log|2\pi K_{\rho,l}|
\end{aligned} \tag{8}
$$

where $S = Z^\top Z + \sigma^2 I$.

The largest rank of matrices in $\log p(\mathbf{u}|\mathbf{y}, X, \boldsymbol{\theta})$ is upper bounded by $\min(rank(C_s), rank(C_f))$. Therefore when either is sparse enough, the entire computation would be fast and efficient.

The first derivative of $\log p(\mathbf{u}|\mathbf{y}, X, \boldsymbol{\theta})$ with regard to $\mathbf{u}$ is given by:

$$
\begin{aligned}
\frac{\partial}{\partial \mathbf{u}_i}\log p(\mathbf{u}|\mathbf{y}, X, \boldsymbol{\theta}) &= -\frac{1}{\sigma^2}\mathrm{Tr}[(S^{-1}Z^\top \mathbf{y}\mathbf{y}^\top Z S^{-1}Z^\top - S^{-1}Z^\top \mathbf{y}\mathbf{y}^\top \\
&\quad +\sigma^2 S^{-1}Z^\top)\frac{\partial}{\partial \mathbf{u}_i}Z] - [K_{\rho,l}^{-1}(\mathbf{u}-b\mathbf{1})]_i
\end{aligned} \tag{9}
$$

The first derivative of $\log p(\mathbf{y}|X, \boldsymbol{\theta}, \boldsymbol{\mu_u})$ with regard to $\boldsymbol{\theta}$ is given by:

$$
\begin{aligned}
\frac{\partial}{\partial \boldsymbol{\theta}_f}\log p(\mathbf{y}|X, \boldsymbol{\theta}) &= -\frac{1}{\sigma^2}\mathrm{Tr}[(S^{-1}Z^\top \mathbf{y}\mathbf{y}^\top Z S^{-1}Z^\top - S^{-1}Z^\top \mathbf{y}\mathbf{y}^\top \\
&\quad +\sigma^2 S^{-1}Z^\top)\frac{\partial}{\partial \boldsymbol{\theta}_f}Z]|_{\mathbf{u}=\boldsymbol{\mu_u}}
\end{aligned} \tag{10}
$$

$$
\begin{aligned}
\frac{\partial}{\partial \sigma^2}\log p(\mathbf{y}|X, \boldsymbol{\theta}) &= -\frac{1}{2}\mathrm{Tr}[\frac{1}{\sigma^4}\mathbf{y}^\top(ZS^{-1}Z^\top - I)\mathbf{y} + \frac{1}{\sigma^2}\mathbf{y}^\top Z S^{-1}S^{-1}Z^\top \mathbf{y} \\
&\quad +S^{-1}]|_{\mathbf{u}=\boldsymbol{\mu_u}}
\end{aligned} \tag{11}
$$

$$\frac{\partial}{\partial b}\log p(\mathbf{y}|X, \boldsymbol{\theta}) = \mathbf{1}^\top K^{-1}(\mathbf{u}-b\mathbf{1})|_{\mathbf{u}=\boldsymbol{\mu_u}} \tag{12}$$

$$
\begin{aligned}
\frac{\partial}{\partial \rho/\partial l}\log p(\mathbf{y}|X, \boldsymbol{\theta}) &= \frac{1}{2}(\mathbf{u}-b\mathbf{1})^\top K^{-1}\left(\frac{\partial}{\partial \rho/\partial l}K\right)K^{-1}(\mathbf{u}-b\mathbf{1}) \\
&\quad -\frac{1}{2}\mathrm{Tr}[K^{-1}\frac{\partial}{\partial \rho/\partial l}K]|_{\mathbf{u}=\boldsymbol{\mu_u}}
\end{aligned} \tag{13}
$$

where

$$\frac{\partial}{\partial \mathbf{u}_i}Z = X\left(\frac{\partial}{\partial \mathbf{u}_i}C_s^{\frac{1}{2}}\right)B^\top C_f^{\frac{1}{2}}B \tag{14}$$

$$\frac{\partial}{\partial \boldsymbol{\theta}_f}Z = XC_s^{\frac{1}{2}}B^\top\left(\frac{\partial}{\partial \boldsymbol{\theta}_f}C_f^{\frac{1}{2}}\right)B|_{\mathbf{u}=\boldsymbol{\mu_u}} \tag{15}$$

We also compute the Hessian matrix of $\log p(\mathbf{u}|\mathbf{y}, X, \boldsymbol{\theta})$ for $\Lambda_{\mathbf{u}}$. The second derivative with regard to $\mathbf{u}$ is given by:

$$\frac{\partial^2}{\partial \mathbf{u}_i \partial \mathbf{u}_j}\log p(\mathbf{u}|\mathbf{y}, X, \boldsymbol{\theta}) = H_a + H_b + H_c + H_d - K^{-1} = -\Lambda_{\mathbf{u}}^{-1}, \tag{16}$$

$$H_a = \frac{1}{\sigma^2}\mathrm{Tr}\left[-(S^{-1}Z^\top \mathbf{y}\mathbf{y}^\top Z S^{-1}Z^\top - S^{-1}Z^\top \mathbf{y}\mathbf{y}^\top + \sigma^2 S^{-1}Z^\top)\left(\frac{\partial^2}{\partial \mathbf{u}_i \partial \mathbf{u}_j}Z\right)\right], \tag{17}$$

$$
\begin{aligned}
H_b \;=\;& \frac{1}{2\sigma^2}\mathrm{Tr}[S^{-1}\left(\frac{\partial}{\partial \mathbf{u}_i}Z\right)^{\top}\mathbf{y}\mathbf{y}^{\top}\left(\frac{\partial}{\partial \mathbf{u}_j}Z\right) + S^{-1}\left(\frac{\partial}{\partial \mathbf{u}_j}Z\right)^{\top}\mathbf{y}\mathbf{y}^{\top}\left(\frac{\partial}{\partial \mathbf{u}_i}Z\right) \\
& -S^{-1}Z^{\top}\left(\frac{\partial}{\partial \mathbf{u}_i}Z\right)S^{-1}\left(\frac{\partial}{\partial \mathbf{u}_j}Z\right)^{\top}\mathbf{y}\mathbf{y}^{\top}Z - S^{-1}\left(\frac{\partial}{\partial \mathbf{u}_i}Z\right)^{\top}ZS^{-1}\left(\frac{\partial}{\partial \mathbf{u}_j}Z\right)^{\top}\mathbf{y}\mathbf{y}^{\top}Z \\
& -S^{-1}Z^{\top}\left(\frac{\partial}{\partial \mathbf{u}_i}Z\right)S^{-1}Z^{\top}\mathbf{y}\mathbf{y}^{\top}\left(\frac{\partial}{\partial \mathbf{u}_j}Z\right) \\
& -S^{-1}\left(\frac{\partial}{\partial \mathbf{u}_i}Z\right)^{\top}ZS^{-1}Z^{\top}\mathbf{y}\mathbf{y}^{\top}\left(\frac{\partial}{\partial \mathbf{u}_j}Z\right)]
\end{aligned}
\tag{18}
$$

$$
\begin{aligned}
H_c \;=\;& \frac{1}{2\sigma^2}\mathrm{Tr}[S^{-1}Z^{\top}\left(\frac{\partial}{\partial \mathbf{u}_i}Z\right)S^{-1}Z^{\top}\mathbf{y}\mathbf{y}^{\top}ZS^{-1}Z^{\top}\left(\frac{\partial}{\partial \mathbf{u}_j}Z\right) \\
& +S^{-1}\left(\frac{\partial}{\partial \mathbf{u}_i}Z\right)^{\top}ZS^{-1}Z^{\top}\mathbf{y}\mathbf{y}^{\top}ZS^{-1}Z^{\top}\left(\frac{\partial}{\partial \mathbf{u}_j}Z\right) \\
& +S^{-1}Z^{\top}\left(\frac{\partial}{\partial \mathbf{u}_i}Z\right)S^{-1}Z^{\top}\mathbf{y}\mathbf{y}^{\top}ZS^{-1}\left(\frac{\partial}{\partial \mathbf{u}_j}Z\right)^{\top}Z \\
& +S^{-1}\left(\frac{\partial}{\partial \mathbf{u}_i}Z\right)^{\top}ZS^{-1}Z^{\top}\mathbf{y}\mathbf{y}^{\top}ZS^{-1}\left(\frac{\partial}{\partial \mathbf{u}_j}Z\right)^{\top}Z \\
& -S^{-1}\left(\frac{\partial}{\partial \mathbf{u}_i}Z\right)^{\top}\mathbf{y}\mathbf{y}^{\top}ZS^{-1}Z^{\top}\left(\frac{\partial}{\partial \mathbf{u}_j}Z\right) - S^{-1}Z^{\top}\mathbf{y}\mathbf{y}^{\top}\left(\frac{\partial}{\partial \mathbf{u}_i}Z\right)S^{-1}Z^{\top}\left(\frac{\partial}{\partial \mathbf{u}_j}Z\right) \\
& -S^{-1}\left(\frac{\partial}{\partial \mathbf{u}_i}Z\right)^{\top}\mathbf{y}\mathbf{y}^{\top}ZS^{-1}\left(\frac{\partial}{\partial \mathbf{u}_j}Z\right)^{\top}Z - S^{-1}Z^{\top}\mathbf{y}\mathbf{y}^{\top}\left(\frac{\partial}{\partial \mathbf{u}_i}Z\right)S^{-1}\left(\frac{\partial}{\partial \mathbf{u}_j}Z\right)^{\top}Z \\
& -S^{-1}Z^{\top}\mathbf{y}\mathbf{y}^{\top}ZS^{-1}\left(\frac{\partial}{\partial \mathbf{u}_i}Z\right)^{\top}\left(\frac{\partial}{\partial \mathbf{u}_j}Z\right) - S^{-1}Z^{\top}\mathbf{y}\mathbf{y}^{\top}ZS^{-1}\left(\frac{\partial}{\partial \mathbf{u}_j}Z\right)^{\top}\left(\frac{\partial}{\partial \mathbf{u}_i}Z\right) \\
& +S^{-1}Z^{\top}\mathbf{y}\mathbf{y}^{\top}ZS^{-1}Z^{\top}\left(\frac{\partial}{\partial \mathbf{u}_i}Z\right)S^{-1}Z^{\top}\left(\frac{\partial}{\partial \mathbf{u}_j}Z\right) \\
& +S^{-1}Z^{\top}\mathbf{y}\mathbf{y}^{\top}ZS^{-1}\left(\frac{\partial}{\partial \mathbf{u}_i}Z\right)^{\top}ZS^{-1}Z^{\top}\left(\frac{\partial}{\partial \mathbf{u}_j}Z\right) \\
& +S^{-1}Z^{\top}\mathbf{y}\mathbf{y}^{\top}ZS^{-1}Z^{\top}\left(\frac{\partial}{\partial \mathbf{u}_i}Z\right)S^{-1}\left(\frac{\partial}{\partial \mathbf{u}_j}Z\right)^{\top}Z \\
& +S^{-1}Z^{\top}\mathbf{y}\mathbf{y}^{\top}ZS^{-1}\left(\frac{\partial}{\partial \mathbf{u}_i}Z\right)^{\top}ZS^{-1}\left(\frac{\partial}{\partial \mathbf{u}_j}Z\right)^{\top}Z],
\end{aligned}
\tag{19}
$$

$$
\begin{aligned}
H_d \;=\;& \frac{1}{2}\mathrm{Tr}[-S^{-1}\left(\frac{\partial}{\partial \mathbf{u}_i}Z\right)^{\top}\left(\frac{\partial}{\partial \mathbf{u}_j}Z\right) - S^{-1}\left(\frac{\partial}{\partial \mathbf{u}_j}Z\right)^{\top}\left(\frac{\partial}{\partial \mathbf{u}_i}Z\right) \\
& +S^{-1}Z^{\top}\left(\frac{\partial}{\partial \mathbf{u}_i}Z\right)S^{-1}\left(\frac{\partial}{\partial \mathbf{u}_j}Z\right)^{\top}Z + S^{-1}\left(\frac{\partial}{\partial \mathbf{u}_i}Z\right)^{\top}ZS^{-1}\left(\frac{\partial}{\partial \mathbf{u}_j}Z\right)^{\top}Z \\
& +S^{-1}Z^{\top}\left(\frac{\partial}{\partial \mathbf{u}_i}Z\right)S^{-1}Z^{\top}\left(\frac{\partial}{\partial \mathbf{u}_j}Z\right) + S^{-1}\left(\frac{\partial}{\partial \mathbf{u}_i}Z\right)^{\top}ZS^{-1}Z^{\top}\left(\frac{\partial}{\partial \mathbf{u}_j}Z\right)]
\end{aligned}
\tag{20}
$$

where

$$
\frac{\partial^2}{\partial \mathbf{u}_i \partial \mathbf{u}_j}Z \;=\; X\left(\frac{\partial^2}{\partial \mathbf{u}_i \partial \mathbf{u}_j}C_s^{\frac{1}{2}}\right)B^{\top}C_f^{\frac{1}{2}}B
\tag{21}
$$

## References

[1] Francis R Bach and Michael I Jordan. Kernel independent component analysis. *The Journal of Machine Learning Research*, 3:1–48, 2003.