[Reviews · NeurIPS 2014]

Submitted by Assigned_Reviewer_2

In their paper the authors propose a prior for linear regression which takes unknown spatial or temporal structures in high-dimensional data into account. This is achieved by using a Gaussian process as hyper-prior for the log-variance of the weights, so that neighboring weights have similar priors. This differs from other approaches by using a hierarchical probabilistic model instead of a simple regularization of the weight vector. Maximum-a-posteriori estimates for the hyper-parameters are calculated numerically, while inference of the weights given the hyper-parameters is possible analytically (Gaussian process regression).

Results on simulated data show that the model is able to deal with spatial structures in the weight vector successfully. The authors also applied their algorithm to fMRI data, where spatial correlations between weights can be expected. Here cross-validation shows improvements compared to standard methods.

The paper is well written and describes the proposed model using basic probabilities, which should be familiar to all readers. The improvements achieved by this method are well demonstrated on both simulated and real data. Additionally, the inference algorithm is quite general and could be applied easily to other high-dimensional regression problems.

However, some notations only become clear by reading the supplementary material. For example, L in equation 10 is not defined in the main paper. Or theta_s is mentioned, but never defined. Is it equal to theta_u in the supplementary material? C_s seems to have a different definition, too. Therefore I recommend that the authors include the relevant definitions in the main paper to make it self-contained.

In their response the authors propose to change the title of the paper "Sparse Dependent Bayesian Structure Learning" to "Sparse Dependent Bayesian Structure Learning with Gaussian Process Priors". In my opinion this change is a useful improvement, as it makes the content more visible.
Summary: Good paper with detailed information on the proposed inference algorithm. Results show visible improvements compared to baseline methods.

Submitted by Assigned_Reviewer_14

I appreciate the thorough author feedback and find the authors' arguments mostly compelling. I have decided to increase my review score to be above the acceptance threshold under the understanding that the authors will clarify the technical concerns in a final paper and expand on the bibliography.

I am still not convinced that the covariance function is positive semi-definite for inputs with dimension greater than one. I understand that as the inputs correspond to the position of the weights they will generally will be of low dimension. However, it is much more interesting to allow the weights to be located in 2D (images), 3D (videos) or even a higher dimensional latent space. As a simple remedy, the authors could make the covariance be the product over dimensions as in [11] or they could restrict the dimensionality of the inputs to be one. Otherwise, I would like more justification in the manuscript or in an appendix that the proposed covariance is PSD.

------------------

This paper explores the problem of linear regression in the scenario where many of the weights are expected to be zero. The innovation of the paper is the development of a hierarchical Bayesian model for which this sparsity follows a spatial pattern, the sparsity of the weights is dependent on the relative spatial location of each weight. The sparsity pattern is achieved through placing a Gaussian process prior on the regression weights. Specifically, the authors place a GP prior, conditioned on the spatial location of the weights, over a latent variable governing the diagonal of a covariance matrix. This is composed with a smoothness covariance in such a way that the latent variables govern the magnitude of the eigenvalues. After fitting the hyperparameters of the model, the authors prune the weights corresponding to small eigenvalues of the covariance, achieving the desired sparsity.

The paper is well written, structured and easy to understand. The authors do a good job of motivating the problem and sparse regression is certainly a topic of interest to the NIPS community. The model seems novel and interesting and the experiments, although very limited, do demonstrate the quality of the approach. There are, however, some concerns with the methodology and the correctness of the model and the empirical evaluation is not exhaustive enough to be compelling.

Although the authors start with a neat model, the approach seems to turn into a rather heuristic procedure due to scalability, tractability and numerical issues. Although I empathize with the authors, the Bayesian model becomes an approximation to an approximation of the original model. The paper seems like a very good start to an interesting model, but the approach seems a little suspect (i.e. PSD-ness, pruning weights corresponding to small eigenvalues, inverting with the SVD due to a singular covariance, various approximations) and thus I would like significantly more justification of this method both in the text and empirically.

A major concern is the chosen covariance function (equation 5). Specifically, I doubt that it is positive definite and I find it odd in general. Why would one have a periodic covariance over the location of the weights? This seems strange as some closer weights will be anticorrelated while farther weights will be correlated. I think this is only positive definite if the distance between the location of the weights i and j is bounded between [pi/2, -pi/2]. Perhaps this is a reason for the numerical instability that is referred to on line 208? The authors must justify that this covariance function is a reasonable choice (i.e. show that it is PSD) and provide some reasoning.

The authors should expand significantly on the related work and references. There is a wealth of literature about structured sparse regression from recent work in submodularity and convex optimization (see e.g. Bach, http://arxiv.org/abs/1111.6453), and in the machine learning literature in general (e.g. Huang et. al http://jmlr.org/papers/volume12/huang11b/huang11b.pdf). There is plenty of space on the references page to add these.

Radford Neal and David Mackay should really be referenced for automatic relevance determination (ARD).

In the abstract, the authors suggest that the Gaussian process prior is conditioned on the Fourier coefficients of the weights. I'm not sure I understand how the Fourier coefficients come into play here and would appreciate some clarification.

There is no analysis done on the impact of the various approximations on the posterior of the model. Specifically, how is the predictive variance of the model affected with respect to the approximations? Do they negatively affect the uncertainty of the model (I think they probably do significantly).

I find it strange that the smooth RVM is left out of the final experiment. This should be included to make the empirical analysis compelling. However, at least some of the wealth of existing structured sparsity work should also be compared to. Even if that work is not Bayesian, it give a better perspective on how difficult the problem is. Finally, it seems to me that Gaussian processes should really be included in the comparison. Why are these left out? It doesn't seem that the proposed model scales beyond what a GP can model.

Summary: The paper is well written, structured and easy to understand. The authors do a good job of motivating the problem and structured sparse regression is certainly a topic of interest to the NIPS community. The model seems novel and the experiments, although very limited, do demonstrate the quality of the approach. There are, however, some significant concerns with the methodology and the correctness of the model and the empirical evaluation is not exhaustive enough to be compelling.

Submitted by Assigned_Reviewer_17

This paper proposes DRD, Dependent Relevance Determination, that uses Gaussian process as priors of covariance to learn smooth and region-sparse weight tensors in regression problems. It can also impose smoothness by using the power spectrum duals that prune high frequencies and encourage sparseness in two bases simultaneously. The model is an improvement over the previous work of ALD (automatic locality determination), which also captures both sparsity and smoonthness as the proposed model does, but ALD can only detect one non-zero region (cluster) while the proposed one can capture more than one.

The model is well illustrated in simulation study to highlight its performance difference from previous methods (Figure 2, 3, 4).

Figure 5: It was hard to notice the difference across images when printed on a paper while it’s noticeable on screen. The authors may consider redrawing the figures, e.g. by drawing the difference between the estimated and the true covariance instead of the raw matrix value. Also, the label for the last column is written as “DRDsf”, but it’s more consistent to write it as “sDRD” since DRDsf does not seem to appear in the main text.

Figure 6. The label used in the main text for the last column of Figure 6 is “DRD” while in the actual figure the label is “sDRD”. Please clarify.

References [6] and [10] seem to be identical ones.

Line 212 (p.4) “so we we apply”
Line 303 ~ 305 (p.6): upper → left, lower → right (regarding Figure 4)

The title is not very informative. The authors may consider changing the title to be more specific.
Summary: The paper presents an improved algorithm over existing work, and the model is well described and validated. There are some minor issues to be addressed as described in comments to authors.
Author Feedback
Author rebuttal: We thank the reviewers for their careful reading of our paper and very constructive set of comments.

Ultimately, it seems the reviewers like the main idea in our paper -- a new, tractable Bayesian approach for capturing dependencies underlying sparse coefficients -- but were put off by lack of clarity in the exposition and concerns about some of the technical details. We felt these reviews were extremely fair, and agree with the vast majority of the points raised.

We feel that we can address these concerns very carefully and thoroughly in the revision, and will try our best to convince the reviewers of soundness on technical points in our remarks below. We hope that this will lift the reviewers’ opinion of the contribution enough to consider placing the paper above the threshold for acceptance to the meeting (with two scores above threshold and one below, it seems that the paper currently sits very close to the border). We are grateful for the reviewers’ help in improving the paper.

== Reviewer 1 ==

Thanks, first for a very clear and articulate summary of our paper. The reviewer seems to have clearly understood the approach and our intended contribution.

We feel that the reviewer’s objections mostly involve technical issues that we did not explain or justify clearly enough, but which do not undermine the novelty or soundness of the basic results. We apologize for not explaining / justifying these issues more carefully, and feel they will be straightforward to address in the revision. We thank the reviewer for raising them.

> positive definiteness of the covariance function (eq. 5), and “odd in general”:
First: the covariance function is indeed positive definite -- we apologize for not motivating it or explaining its origin more clearly. Positive definiteness can be proven using Bochner’s theorem (see [11], which discusses this particular form explicitly in eqs. 6-11). Note that the restriction to [pi/2, -pi/2] ensures only element-wise positivity of the kernel, which does not suffice to establish positive-definiteness of the kernel itself. So, that restriction is not necessary for our kernel and there is no numerical issues regarding this point.

Second: the justification for this choice is that it confers the ability to detect spatial or temporal structure in particular frequency bands. The squared-exponential kernel always gives more variance to lower frequencies, but in some datasets we may find patterns of sparse coefficients that are dominated by Fourier modes with intermediate frequencies. This covariance function allows such patterns to emerge while suppressing higher and lower frequencies. A secondary (numerical) consideration underlying this choice is the availability of an analytic eigendecomposition (into Fourier modes with all-positive weights), which allows for very rapid and accurate determination of a low-rank approximation to the Gram matrix given the hyperparameters. We will explain this more clearly in the revision.

> how Fourier coefficients come into play in the prior:
We apologize for not making this clearer. Our intention is to restrict the high power regions in Fourier domain around low frequency, since the inverse Fourier transform of that form will yield a smooth signal in time domain. The effects are similar to the DFT approach employed by [6], with the advantage of significantly improved computational performance and scalability.

> impact of various approximations on the posterior:
Various computational tricks were applied to significantly improve the scalability of the proposed approach. We address each in turn with justification:
(1) Pruning weights corresponding to small values of its prior covariance: We note that such pruning is a standard technique applied only to entries that are significantly close to zero. We have analyzed the effect of such pruning, and (provided the eigenvalue cutoff is low enough, as it was here), it has no numerically detectable effect the model performance.
(2) Kernel inversion with the SVD: We found the eigen-decomposition approach resulted in more stable solutions when a few of the kernel eigenvalues were small. Further, we note that the computational cost of matrix inversion and eigen-decomposition are comparable (both cubic in matrix size).
(3) Approximate inference via. Laplace approximation: it’s true that the Laplace approximation to the posterior over the sparsity-promoting latent variables “u” may introduce discrepancies between the true and approximate evidence. We employ it primarily because of the speed/tractability it confers, but we can examine how critical the approximation is by comparing to sampling-based inference (which is tractable in this setting because of the Gaussian prior and analytically tractable marginal likelihood). However, we note that the proposed approach significantly outperformed baseline models on both simulated data (where ground truth is known) and in held-out test data, indicating that the performance is very good even with this approximation.

> Expand significantly on related work/references:
Thank you, we apologize for the numerous missing refs, and will expand those substantially. Thanks again for all these suggestions.

== Reviewer 2 ==

> Visualization of Figure 5, naming “DRDsf”:
Thanks for the suggestion. We will up-sample the figure in the final version for better visualization.

> Typos and not very informative title:
Thank you for pointing those out. We will correct errors. Also, we propose changing the title to ’Sparse Dependent Bayesian Structure Learning with Gaussian Process Priors’, although we would also be happy to take suggestions from the reviewers if they can think of a title that would be more informative.

== Reviewer 3 ==

We thank the reviewer for the positive feedback.

> notational mismatch in manuscript and appendix:
We apologize for the confusion, and will take care to unify all notation in the final version.